# Outcomes and Predictors of 30-Day Readmission in Patients with Hepatocellular Carcinoma Undergoing Transarterial Chemoembolization between 2016 and 2018

Ifrah Fatima [1,*] , Mohamed Ahmed [1] , Wael T. Mohamed [1] , Vinay Jahagirdar [1] , Kevin F. Kennedy [2] and Alisa Likhitsup [3,4]

1   Department of Internal Medicine, University Health Truman Medical Center, University of Missouri, 2301 Holmes Street, Kansas City, MO 64108, USA; m.ahmed@umkc.edu (M.A.); jahagirdarv@umkc.edu (V.J.)
2   Department of Cardiovascular Research, Saint Luke's Hospital, Kansas City, MO 64111, USA; kfkennedy@saint-lukes.org
3   Department of Gastroenterology, University of Missouri, Kansas City, MO 64111, USA; allikhit@med.umich.edu
4   Saint Luke's Liver Disease and Transplant Specialists, Saint Luke's Hospital, Kansas City, MO 64111, USA
*   Correspondence: ifrahfatima@umkc.edu; Tel.: +1-816-404-0950

**Abstract:** Background: Hepatocellular carcinoma (HCC) is the third leading cause of cancer death worldwide. The 5-year survival rate for liver cancer in the US has improved from 3% four decades ago to 20% now. Transarterial chemoembolization (TACE) is the treatment of choice for stage B/intermediate-stage HCC. Complications of TACE include hepatic encephalopathy, liver failure, post-embolization syndrome, duodenal ulcers, liver abscesses, acute cholecystitis, and injury to the biliary tract. This study evaluates the 30-day readmission rate and predictors of readmission among patients with HCC undergoing TACE. Methods: The 2016–2018 Healthcare Cost and Utilization Project (HCUP) database, which includes the National Readmission Database (NRD), was used. All adult patients with HCC who underwent TACE were identified using the International Classification of Diseases (ICD-10). The rate of 30-day readmissions after TACE and the associated diagnoses were identified. Logistic regression was used to obtain adjusted odds ratios for variables associated with 30-day readmission. Results: A total of 566 patients underwent TACE between 2016–2018. Sixty-five patients were excluded due to death and unavailability of 30-day readmission data. The procedure was performed in large (80.4%), metro-teaching hospitals (94.5%). Mean patient age was 65.1 ± 9.9 years, and 74% of patients were male. Among the 501 patients, 81 (16.2%) were readmitted within 30 days. The mean age for readmitted patients was 63.2 ± 11.0 and 69.1% were male. The mean length of stay at readmission was 5.5 ± 7.3 days. A total of 7.4% of patients had neurological disorders, 17.3% had weight loss, 30.9% had fluid and electrolyte imbalance, and 21.0% had hepatic encephalopathy. The most common primary diagnoses at 30-day readmission were liver cell carcinoma, sepsis, and liver failure. Univariate analysis for variables associated with 30-day readmission included hepatic encephalopathy (OR 3.45; 95% CI 1.8–6.62; *p* = 0.0002), underlying neurological disorders (OR 3.28; 95% CI 1.16–9.3; *p* = 0.03), weight loss (OR 2.82; 95% CI 1.42–5.61; *p* = 0.003), and Medicaid status (OR 1.74; 95% CI 1.05–2.88; *p* = 0.03). Multivariable analysis showed hepatic encephalopathy (OR 2.91; 95% CI 1.4, 6.04; *p* = 0.04) and weight loss (OR 2.37; 95% CI 1.13–4.96; *p* = 0.02) were associated with hospital readmission. Conclusions: Weight loss and hepatic encephalopathy were predictors for 30-day readmission after a TACE procedure for HCC.

**Keywords:** transarterial chemoembolization; hepatocellular carcinoma; readmission; hepatic encephalopathy; weight loss; cirrhosis

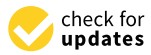



## 1. Introduction

Hepatocellular carcinoma (HCC) is the third leading cause of cancer death worldwide. The general 5-year survival rate for liver cancer in the United States has improved to 21%,

compared to 3% about 40 years ago [1]. HCC patients are classified using the Barcelona Clinic Liver Cancer (BCLC) staging system, which considers the tumor number and size, performance status, and liver function (Child–Pugh–Turcotte score). Transcatheter arterial chemoembolization (TACE) is the treatment of choice for stage B/intermediate-stage HCC [2].

TACE initially evolved in the early 1980s as a therapy for HCC. It uses chemotherapy agents to embolize the hepatic artery, which is the main supply of the tumor. The most common side effects of TACE, despite being regarded as a safe procedure, are acute cholecystitis, leukopenia, and post-embolization syndrome [3]. Other reported complications are pulmonary thromboembolism, hepatic ischemia, liver abscess, bile duct lesions, and acute pancreatitis [4–7]. Some of the more serious complications, like spontaneous rupture of liver cancer, duodenal perforation, liver abscesses, and hepatic artery occlusion, are rare [8].

Two techniques of TACE are in practice: conventional TACE (cTACE) and TACE with drug-eluting beads (DEB-TACE). cTACE uses a lipiodol-based emulsion with an embolizing agent to ensure the transcatheter delivery of the chemotherapeutic agents. DEB-TACE causes a slow release of chemotherapeutic agents, resulting in an increased intensity and duration of ischemia. TACE uses two strategies to achieve its favorable effects. First, since the hepatic artery supplies blood to most hepatic tumors, arterial embolization cuts off this supply, delaying growth until neovascularity replaces it. Second, focused chemotherapy administration increases the dose delivered to the tissue while simultaneously lowering systemic exposure, which is usually the dose-limiting factor. The cytotoxic effect on the tumor cells is increased and the side effects of the chemotherapy drugs are decreased with a high drug concentration in the tumor area. The chemotherapeutic medication is not removed from the tumor bed after embolization and this effect is amplified [7–9]. Hospital readmissions are an important quality-of-care measure that have been monitored for improvement. The NRD is the largest public readmission database developed as a part of the Healthcare Cost and Utilization Project regardless of the expected payer for the hospital stay. We aimed to evaluate the 30-day readmission rate and predictors of readmissions among patients with HCC undergoing TACE.

## 2. Methods

### 2.1. Data Source

The 2016–2018 Healthcare Cost and Utilization Project (HCUP) database that includes the National Readmission Database (NRD) was used. It is the country's largest government-initiated group of databases and includes all-payer and encounter-level hospital care data. It accounts for about 36 million weighted discharges per year, with contributions from 27 states in the year 2016. It was developed through a federal–state–industry partnership and sponsored by the Agency for Healthcare Research and Quality (AHRQ). The NRD is the HCUP database that focuses on hospital readmissions using verified, deidentified, and linked patient data from HCUP state inpatient databases. This stratified probability sample represents all non-federal acute care inpatient hospitalizations in the country. It encompasses a random 20% sample of all discharged patients from the participating hospitals within each category (like ownership/control, bed size, teaching status, urban/rural location, and geographic region) [10].

### 2.2. Study Population

All adult patients with hepatocellular carcinoma who underwent TACE between 2016 and 2018 were included in this study. Patients with liver malignancy were identified using the International Classification of Diseases (ICD-10) [ICD-10 codes C22.0, C22.8, C22.9]. The primary procedure of interest was TACE, which was the primary exposure variable. It was identified using the CPT codes 04L33DZ (Occlusion of Hepatic Artery with Intraluminal Device, Percutaneous Approach) and 3E05305 (Introduction of Other Antineoplastic into Peripheral Artery, Percutaneous Approach). A total of 566 patients who underwent TACE were identified and 65 patients who died during the initial hospitalization were excluded.

*2.3. Statistical Analysis*

The rate of 30-day readmission after TACE and the associated diagnoses were identified. We compared the baseline characteristics of each outcome to identify the factors associated with hospital readmission using the Wilcoxon rank-sum test, chi-square, or Fisher's exact test. We used univariable logistic regression to calculate unadjusted odds ratios and 95% confidence intervals and included variables reaching a significance level of $p < 0.20$ in a multivariable logistic regression model. Two-tailed $p$-values below 0.05 were considered statistically significant. All analyses were conducted in Stata 13 (StataCorp, College Station, TX, USA).

## 3. Results

*3.1. Patient Characteristics*

A total of 501 patients were included in this study. A large proportion of these were admitted in the year 2018 (42%) and at large (80.4%) and metro-teaching hospitals (94.5%). A total of 78.6% of these were at private and non-profit hospitals and only 18.2% at government-owned hospitals. The mean age in the TACE population was $65.1 \pm 9.9$ years. A large proportion of the procedures were performed in males (74%). Patients were assessed for the presence of complications and were found to have liver disease (75.4%), hypertension (60.2%), diabetes without chronic complications (38.7%), coagulopathy (23.1%), and fluid and electrolyte disorders (20.1%).

Of the total 501 patients, 81 patients (16%) were readmitted in 30 days. Readmission rates were highest in 2018 (44.4%), followed by 2017 (30.9%) and 2016 (24.7%). This corresponded with the number of TACE procedures performed through the years. The mean age in this cohort was slightly lower than in the patients who were not readmitted ($63.2 \pm 11.0$ vs. $65.5 \pm 9.2$; $p < 0.045$). The length of hospital stay was longer among the readmitted group ($5.5 \pm 7.3$ vs. $3.7 \pm 6.0$; $p < 0.02$). The mean of the total charges incurred by the patients and hospitals was calculated and there was no significant difference seen in the readmission group (USD $109,897.7 \pm 82,243.8$ vs. USD $98,221.0 \pm 88,644.7$; $p = 0.27$). There was no significant difference between the insurance status of the readmitted population and that of the patients who did not require readmission. The teaching status of urban hospitals as well as the control/ownership of the hospitals were comparable in the two groups (Table 1).

**Table 1.** Baseline characteristics.

| | Total | 30-Day | | $p$-Value |
|---|---|---|---|---|
| | | **Non-Elective Readmission** | | |
| | $n = 566$ | $n = 81$ | $n = 420$ | |
| Calendar year | | | | |
| 2016 | 174 (30.7%) | 20 (24.7%) | 139 (33.1%) | 0.329 |
| 2017 | 154 (27.2%) | 25 (30.9%) | 117 (27.9%) | |
| 2018 | 238 (42.0%) | 36 (44.4%) | 164 (39.0%) | |
| Age in years at admission | | | | |
| Mean $\pm$ SD | $65.1 \pm 9.9$ | $63.2 \pm 11.0$ | $65.5 \pm 9.2$ | 0.045 |
| Median (IQR) | 65.0 (60.0, 71.0) | 63.0 (57.0, 71.0) | 65.0 (60.0, 71.0) | |
| Female | 147 (26.0%) | 25 (30.9%) | 109 (26.0%) | 0.360 |
| Length of stay | | | | |
| Mean $\pm$ SD | $4.0 \pm 6.1$ | $5.5 \pm 7.3$ | $3.7 \pm 6.0$ | 0.015 |
| Median (IQR) | 2.0 (1.0, 4.0) | 2.0 (1.0, 7.0) | 2.0 (1.0, 4.0) | |

**Table 1.** *Cont.*

| | Total | 30-Day | | *p*-Value |
|---|---|---|---|---|
| | | **Non-Elective Readmission** | | |
| | ***n* = 566** | ***n* = 81** | ***n* = 420** | |
| Primary expected payer | | | | |
| Medicare | 303 (53.5%) | 34 (42.0%) | 230 (54.8%) | |
| Medicaid | 148 (26.1%) | 29 (35.8%) | 102 (24.3%) | |
| Private insurance | 92 (16.3%) | 16 (19.8%) | 69 (16.4%) | 0.216 |
| Self-pay | 5 (0.9%) | 0 (0.0%) | 4 (1.0%) | |
| No charge | 1 (0.2%) | 0 (0.0%) | 1 (0.2%) | |
| Other | 17 (3.0%) | 2 (2.5%) | 14 (3.3%) | |
| Patient Location: NCHS Urban–Rural Code | | | | |
| 1 Central counties of metro areas of ≥1 million population | 248 (44.4%) | 40 (50.0%) | 183 (44.2%) | |
| 2 Fringe counties of metro areas of ≥1 million population | 120 (21.5%) | 17 (21.3%) | 90 (21.7%) | |
| 3 Counties in metro areas of 250,000–999,999 population | 110 (19.7%) | 16 (20.0%) | 78 (18.8%) | 0.499 |
| 4 Counties in metro areas of 50,000–249,999 population | 31 (5.5%) | 3 (3.8%) | 24 (5.8%) | |
| 5 Micropolitan counties | 31 (5.5%) | 1 (1.3%) | 26 (6.3%) | |
| 6 Not metropolitan or micropolitan counties | 19 (3.4%) | 3 (3.8%) | 13 (3.1%) | |
| Missing | 7 | 1 | 6 | |
| Total charges (cleaned) | | | | |
| Mean ± SD | 102,720.4 ± 88,242.7 | 109,897.7 ± 82,243.8 | 98,221.0 ± 88,644.7 | |
| Median (IQR) | 76,797.0 (50,924.0, 124,164.0) | 88,985.5 (56,834.5, 140,748.0) | 72,118.5 (49,744.0, 117,577.0) | 0.275 |
| Missing | 1 | 1 | 0 | |
| Median household income national quartile for patient ZIP code | | | | |
| 1 | 160 (28.6%) | 24 (30.4%) | 120 (28.9%) | |
| 2 | 143 (25.6%) | 21 (26.6%) | 108 (26.0%) | 0.980 |
| 3 | 136 (24.3%) | 17 (21.5%) | 98 (23.6%) | |
| 4 | 120 (21.5%) | 17 (21.5%) | 89 (21.4%) | |
| Missing | 7 | 2 | 5 | |
| Bed size of hospital | | | | |
| Small | 13 (2.3%) | 2 (2.5%) | 9 (2.1%) | |
| Medium | 98 (17.3%) | 12 (14.8%) | 75 (17.9%) | 0.777 |
| Large | 455 (80.4%) | 67 (82.7%) | 336 (80.0%) | |
| Control/ownership of hospital | | | | |
| Government | 103 (18.2%) | 9 (11.1%) | 86 (20.5%) | |
| Private non-profit | 445 (78.6%) | 70 (86.4%) | 321 (76.4%) | 0.111 |
| Private invest–own | 18 (3.2%) | 2 (2.5%) | 13 (3.1%) | |

| | **Total** | **30-Day** | | *p*-Value |
| | | **Non-Elective Readmission** | | |
| | *n* = 566 | *n* = 81 | *n* = 420 | |
|---|---|---|---|---|
| Teaching status of urban hospitals | | | | |
| Metro non-teaching | 30 (5.3%) | 4 (4.9%) | 20 (4.8%) | 1.000 |
| Metro Teaching | 535 (94.5%) | 77 (95.1%) | 399 (95.0%) | |
| Non-metro | 1 (0.2%) | 0 (0.0%) | 1 (0.2%) | |
| Valvular disease | 14 (2.5%) | 1 (1.2%) | 11 (2.6%) | 0.700 |
| Pulmonary circulation disease | 3 (0.5%) | 1 (1.2%) | 2 (0.5%) | 0.411 |
| Peripheral vascular disease | 35 (6.2%) | 5 (6.2%) | 27 (6.4%) | 0.931 |
| Paralysis | 3 (0.5%) | 1 (1.2%) | 2 (0.5%) | 0.411 |
| Other neurological disorders | 18 (3.2%) | 6 (7.4%) | 10 (2.4%) | 0.030 |
| Chronic pulmonary disease | 103 (18.2%) | 20 (24.7%) | 70 (16.7%) | 0.084 |
| Diabetes without chronic complications | 219 (38.7%) | 28 (34.6%) | 162 (38.6%) | 0.496 |
| Hypothyroidism | 53 (9.4%) | 8 (9.9%) | 38 (9.0%) | 0.813 |
| Renal failure | 59 (10.4%) | 11 (13.6%) | 38 (9.0%) | 0.208 |
| Liver disease | 427 (75.4%) | 59 (72.8%) | 318 (75.7%) | 0.583 |
| Peptic ulcer disease and bleeding | 12 (2.1%) | 3 (3.7%) | 6 (1.4%) | 0.165 |
| Acquired immune deficiency syndrome | 3 (0.5%) | 1 (1.2%) | 2 (0.5%) | 0.411 |
| Rheumatoid arthritis/collagen vascular disorder | 14 (2.5%) | 2 (2.5%) | 8 (1.9%) | 0.668 |
| Coagulopathy | 131 (23.1%) | 24 (29.6%) | 88 (21.0%) | 0.086 |
| Obesity | 68 (12.0%) | 9 (11.1%) | 49 (11.7%) | 0.886 |
| Weight loss | 49 (8.7%) | 14 (17.3%) | 29 (6.9%) | 0.002 |
| Fluid and electrolyte disorders | 114 (20.1%) | 25 (30.9%) | 70 (16.7%) | 0.002 |
| Alcohol abuse | 85 (15.0%) | 13 (16.0%) | 62 (14.8%) | 0.766 |
| Drug abuse | 28 (4.9%) | 5 (6.2%) | 17 (4.0%) | 0.377 |
| Psychosis | 14 (2.5%) | 4 (4.9%) | 9 (2.1%) | 0.240 |
| Depression | 53 (9.4%) | 10 (12.3%) | 37 (8.8%) | 0.317 |
| Hypertension | 341 (60.2%) | 47 (58.0%) | 258 (61.4%) | 0.565 |
| Hepatic encephalopathy | 53 (9.4%) | 17 (21.0%) | 30 (7.1%) | <0.001 |

### 3.2. Diagnosis at Readmission

The various conditions that were identified by ICD codes to record the readmission diagnosis were hypertension, diabetes, renal failure, liver disease, hypothyroidism, chronic pulmonary disease, other neurological disorders, pulmonary circulation disease, valvular disease, paralysis, peptic ulcer disease and bleeding, acquired immune deficiency syndrome, rheumatoid arthritis, coagulopathy, obesity, weight loss, fluid and electrolyte disorders, alcohol use, drug abuse, psychoses, and depression. The presence of weight loss and fluid and electrolyte disturbances was higher in the readmitted population (17.3% vs. 6.9%; $p < 0.01$ and 30.9% vs. 16.7%; $p < 0.01$, respectively). Other significant diagnoses at readmission included neurological disorders (7.4% vs. 2.4%; $p < 0.03$) and hepatic encephalopathy (21% vs. 7.1%; $p < 0.01$).

The primary diagnosis codes at readmission were recorded (Table 2). The most common readmission diagnoses were liver cell carcinoma (9.88%), sepsis (7.41%), hepatic

failure but without coma (4.94%), pulmonary embolism (3.7%), alcoholic cirrhosis without ascites (3.7%), and antineoplastic chemotherapy (3.7%) (Table 2).

**Table 2.** List of most common diagnoses at readmission along with International Classification of Diseases (ICD)-10 Codes.

| ICD-10 Diagnosis Code | Diagnosis | % (*n*) |
| --- | --- | --- |
| C220 | Liver cell carcinoma | 9.88 (8) |
| A419 | Sepsis | 7.41 (6) |
| K7290 | Hepatic failure, unspecified, without coma | 4.94 (4) |
| I2699 | Pulmonary embolism | 3.70 (3) |
| K7031 | Alcoholic cirrhosis with ascites | 3.70 (3) |
| Z5111 | Antineoplastic chemotherapy | 3.70 (3) |
| A4150 | Gram-negative sepsis, unspecified | 2.47 (2) |
| G893 | Neoplasm-related pain | 2.47 (2) |
| K5669 | Other intestinal obstruction | 2.47 (2) |
| K7469 | Other cirrhosis of liver | 2.47 (2) |
| K9189 | Other procedural complications and disorders of digestive system | 2.47 (2) |
| N390 | Urinary tract infection | 2.47 (2) |
| R0789 | Chest pain | 2.47 (2) |
| T8189XA | Other complications of procedures, not elsewhere classified | 2.47 (2) |
| A047 | Enterocolitis due to Clostridium difficile | 1.23 (1) |
| A414 | Sepsis due to anaerobes | 1.23 (1) |
| A4151 | Sepsis due to Escherichia coli | 1.23 (1) |
| A4181 | Sepsis due to Enterococcus | 1.23 (1) |
| B190 | Unspecified viral hepatitis with hepatic coma | 1.23 (1) |
| D735 | Infarction of spleen | 1.23 (1) |
| E1165 | Type 2 diabetes mellitus with hyperglycemia | 1.23 (1) |
| G8918 | Acute post-procedural pain | 1.23 (1) |
| G8929 | Other chronic pain | 1.23 (1) |
| H8112 | Benign paroxysmal vertigo | 1.23 (1) |
| I160 | Hypertensive urgency | 1.23 (1) |
| I481 | Persistent atrial fibrillation | 1.23 (1) |
| I4891 | Unspecified atrial fibrillation | 1.23 (1) |
| I724 | Aneurysm of artery of lower extremity | 1.23 (1) |
| J90 | Pleural effusion | 1.23 (1) |
| K420 | Umbilical hernia with obstruction | 1.23 (1) |
| K439 | Ventral hernia without obstruction | 1.23 (1) |
| K5791 | Diverticulosis of intestine without perforation or abscess | 1.23 (1) |
| K5900 | Constipation | 1.23 (1) |
| K625 | Hemorrhage of anus or rectum | 1.23 (1) |
| K648 | Other hemorrhoids | 1.23 (1) |
| K659 | Peritonitis | 1.23 (1) |
| K72.00 | Acute and subacute hepatic failure without coma | 1.23 (1) |

**Table 2.** *Cont.*

| ICD-10 Diagnosis Code | Diagnosis | % (*n*) |
|---|---|---|
| K7460 | Unspecified cirrhosis of liver | 1.23 (1) |
| K7589 | Other specified inflammatory liver diseases | 1.23 (1) |
| K763 | Infarction of liver | 1.23 (1) |
| K8010 | Calculus of gallbladder with chronic cholecystitis | 1.23 (1) |
| K819 | Cholecystitis | 1.23 (1) |
| K920 | Hematemesis | 1.23 (1) |
| N170 | Acute kidney failure with tubular necrosis | 1.23 (1) |
| R233 | Spontaneous ecchymoses | 1.23 (1) |
| R502 | Drug-induced fever | 1.23 (1) |
| R509 | Fever | 1.23 (1) |

*3.3. Predictors for 30-Day Readmission*

Univariate logistic regression showed the factors associated with 30-day readmission were younger age (OR 0.78; 95% CI 0.61–1; $p$ = 0.04), and Medicaid status (OR 1.74; 95% CI 1.05–2.88; $p$ < 0.03). The readmission diagnoses of underlying neurological disorders (OR 3.28; 95% CI 1.16–9.3; $p$ = 0.03), weight loss (OR 2.82; 95% CI 1.42–5.61; $p$ < 0.01), and hepatic encephalopathy (OR 3.45; 95% CI 1.8–6.62; $p$ < 0.01) were significant for 30-day readmission.

Multivariable logistic regression analysis was performed after adjusting for age, sex, Medicaid status, COPD, other neurological disease, renal failure, PUD, coagulopathy, weight loss, hepatic encephalopathy, and liver failure. The independent predictors associated with 30-day readmission were weight loss (OR 2.37; 95% CI 1.13–4.96; $p$ = 0.02) and hepatic encephalopathy (OR 2.91; 95% CI 1.4, 6.04; $p$ = 0.04) (Table 3). Chronic conditions like lung diseases and neurological disorders trended towards higher odds as well but did not reach statistical significance (OR 1.79; 95% CI 0.99–3.26; $p$ = 0.05 and OR 2.63; 95% CI 0.85–8.12; $p$ = 0.09, respectively). Neither peptic ulcer disease nor presence of coagulopathy were shown to have statistical significance in the measured readmission outcomes.

**Table 3.** Univariate and multivariate analysis for predictors associated with 30-day readmission.

| Label | Univariate OR<br>OR (95% CI) | *p*-Value | Adjusted OR *<br>OR (95% CI) | *p*-Value |
|---|---|---|---|---|
| Age | 0.78 (0.61, 1) | 0.0470 | 0.88 (0.65, 1.18) | 0.3816 |
| Female | 1.27 (0.76, 2.14) | 0.3612 | 1.42 (0.82, 2.48) | 0.2133 |
| Medicaid | 1.74 (1.05, 2.88) | 0.0322 | 1.5 (0.83, 2.72) | 0.1805 |
| Chronic lung disease | 1.64 (0.93, 2.89) | 0.0872 | 1.79 (0.99, 3.26) | 0.0556 |
| Neurological disorders | 3.28 (1.16, 9.3) | 0.0254 | 2.63 (0.85, 8.12) | 0.0922 |
| Renal failure | 1.58 (0.77, 3.24) | 0.2119 | 1.67 (0.76, 3.65) | 0.1983 |
| Peptic ulcer disease | 2.65 (0.65, 10.84) | 0.1739 | 2.81 (0.66, 11.93) | 0.1620 |
| Coagulopathy | 1.59 (0.93, 2.7) | 0.0880 | 1.06 (0.57, 1.96) | 0.8549 |
| Weight loss | 2.82 (1.42, 5.61) | 0.0032 | 2.37 (1.13, 4.96) | 0.0222 |
| Psychosis | 2.37 (0.71, 7.9) | 0.1592 | 2.01 (0.52, 7.72) | 0.3118 |
| Hepatic encephalopathy | 3.45 (1.8, 6.62) | 0.0002 | 2.91 (1.4, 6.04) | 0.0041 |
| Acute hepatic failure without coma | 2.63 (0.47, 14.62) | 0.2684 | 1.43 (0.21, 9.58) | 0.7136 |

* adjusted for age, sex, Medicaid, OPD, other neurological disease, renal failure, peptic ulcer disease, coagulopathy, weight loss, hepatic encephalopathy, and liver failure.

## 4. Discussion

Thirty-day readmission is a significant indicator of the quality of hospital care and services. The United States economic and quality improvement agenda declared the reduction of hospital readmissions as one of its goals [11]. The overall 30-day readmission rate after TACE per our analysis was 16%. This included patients with any non-elective readmission to the hospital. However, there were no data on discharge after an outpatient observation period or on elective readmission.

The readmitted population was slightly younger, and these patients had a longer duration of stay, likely indicating that they were sicker at the first admission itself. Significant diagnoses at readmission included weight loss, fluid and electrolyte disturbances, neurological disorders, and hepatic encephalopathy. Hepatocellular carcinoma and sepsis were the most common primary diagnosis at readmission. Age and Medicaid status were found to be predictive for TACE readmissions in univariate analysis but only weight loss and hepatic encephalopathy were found to be independent predictors after adjusting for various factors mentioned previously.

A recent retrospective observational cohort study by Hund et al. published in March 2023 evaluated the effect of same-day discharge after a 3 h observation period versus discharge after an overnight admission following TACE. The 30-day readmission rates were similar between the groups (4.6%) [12]. Fritshce et al. performed a similar comparison between readmissions after same-day discharge versus overnight observation. The 30-day readmission rate was slightly higher among the same-day discharge (13.8%) group compared to overnight observation (9%) but this was not significant ($p = 0.33$) [13]. Another retrospective analysis from 2016 showed a similar rate of 4.2% over a 21-year period [14]. These rates were much lower than those seen in our study. Our data utilized the NRD, which captures the nationwide population in the US, compared with single-center retrospective data. However, our data were analyzed based on the procedure and diagnosis codes, which are subject to coding inaccuracies due to inaccurate entries or the misclassification of various conditions.

McCarthy et al. reported that Medicare patients were more likely to be readmitted. Our study showed that Medicaid status had significantly increased chances of readmission following TACE, but this was not significant after various other factors were adjusted for.

In our study, the mean age was slightly lower among readmitted patients without statistical significance. One retrospective analysis demonstrated being female was significantly associated with readmission, but this was not seen in our study [14].

Hund et al. also reported that patients rated Child–Turcotte–Pugh (CTP) B or C had a higher rate of readmission (10% vs. 2.9% Child–Pugh B/C vs. A) and more likely to be readmitted within 30 days (OR 2.1; 95% CI 0.5–8.4; $p = 0.04$) [14]. The NRD did not record the CTP scores of the patients, and this was not included in our analysis. We would expect patients with higher CTP class and MELD scores to be sicker and hence to potentially require readmission for various reasons. Chemoembolization is usually avoided in patients with advanced liver failure, including Child-Pugh (CTP) Class C patients. Roth et al. compared patients < 70 and >70 years of age undergoing TACE. They described overall similar rates of rehospitalization after TACE, irrespective of age. In multivariate analysis, a Child score greater than or equal to B7 ($p < 0.01$), an ECOG (Eastern Cooperative Oncology Group) grade greater than or equal to 1 ($p = 0.01$), and a MELD score greater than or equal to 9 ($p = 0.04$) were associated with significant adverse events (worsening of ECOG status, liver decompensation, TACE-related death) [15]. These did not include morbidity and rehospitalizations. It would be interesting to further investigate the CTP and MELD scores of TACE patients and their correlation with outcomes such as readmission and mortality.

Our multivariable analysis showed weight loss and hepatic encephalopathy to significantly correlate with 30-day hospital readmission. An early study from India in 2013 showed that almost 58% patients who were considered for TACE presented with symptoms of weight loss [16]. The impact of critical weight loss (>5%) has been associated with decreased overall survival [17]. This high prevalence of weight loss in the population

undergoing TACE might also be reflected in the diagnosis at readmission that we see in the above data. It is critical to educate patients with cirrhosis regarding a high-protein diet of 1.2–1.5 g/kg per day [18].

The general readmission rates for cirrhosis patients were close to 13%, with hepatic encephalopathy being strongly associated with it [19]. The prevalence of hepatic encephalopathy due to hepatic decompensation can be seen as frequently as 10.8% [19]. Some factors that correlated with decompensation included CTP and MELD scores, initial tumor size, and basal albumin level. However, only pre-TACE bilirubin levels were found to have a predictive value for hepatic encephalopathy after TACE [20]. The finding of hepatic encephalopathy as a predictor for readmission makes it even more important to treat underlying hepatic decompensation in our patients who proceed with TACE. Objective data such as bilirubin levels were not recorded in the database that we queried.

*Limitations*

This study has certain limitations due to the NRD being an administrative database. It is well known that coding inaccuracies arise due to inaccurate entries or the misclassification of various conditions. Although the database represents a larger population than the retrospective studies cited in this article, patient-level data are limited due to the risk of coding errors. There is lack of objective clinical data, including laboratory investigations and imaging, which prevents the assessment of various liver mortality scores, including MELD and CPT, which are known predictors of mortality in patients with cirrhosis. The NRD can only record in-hospital mortality, which might lead to underacknowledged mortality. Some metrics that were not reported include hospital costs, cost to patients, and reimbursement figures, which is a barrier in performing a direct cost analysis of the financial impact of readmissions. Readmission rates have been criticized by many as a poor indicator of quality and have not been conclusively proven to correlate with established quality metrics like mortality or hospital burden. To fully comprehend the connection between readmission rates and quality of care, additional research is necessary. Despite these limitations, this analysis is derived from a large and comprehensive database and highlights important factors associated with readmission after TACE.

## 5. Conclusions

In conclusion, this study offers a large and thorough analysis of readmission rates in the United States following TACE for HCC, providing an understanding of readmission patterns in this particular patient population. Studying the relationship between readmissions and patient factors is important to reduce overall hospital costs. Several factors at the patient level are indicators of readmission, including age, nutritional status, cancer stage, and the severity of the underlying liver disease. Knowledge about the predictors, outcomes, and trends associated with TACE can be useful to clinicians in potentially treating modifiable factors or predicting patients at high risk of readmission and acting on interventions aimed at reducing the risk of hospital readmissions. The relationship between hospital-associated factors and readmissions was less convincing and brings into question the measurement of 30-day readmission data to penalize healthcare systems. We advise future prospective data collection across multiple centers to better identify risk factors and create efficient interventions.

**Author Contributions:** I.F. drafted the manuscript, collected the data, and is the article guarantor. M.A., W.T.M. and V.J. were responsible for the review of the literature and reviewing the article. K.F.K. performed the statistical analysis. A.L. was responsible for the study conception, analysis, critical review, and supervision. All authors have read and agreed to the published version of the manuscript.

**Funding:** This research received no external funding.

**Institutional Review Board Statement:** Ethical review and approval were waived for this study due to publicly available database study.

**Informed Consent Statement:** Not applicable.

**Data Availability Statement:** Data are contained within the article.

**Conflicts of Interest:** The authors declare no conflicts of interest.

## Abbreviations

| | |
|---|---|
| ICD | International Classification of Diseases |
| HCUP | Healthcare Cost and Utilization Project |
| NRD | National Readmission Database |
| TACE | Transarterial Chemoembolization |
| CTP | Child–Turcotte–Pugh |
| HCC | hepatocellular carcinoma |
| ECOG | Eastern Cooperative Oncology Group |

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
