# Peer review of "Outcomes and Predictors of 30-Day Readmission in Patients with Hepatocellular Carcinoma Undergoing Transarterial Chemoembolization between 2016 and 2018"

_gastroent, doi:10.3390/gastroent15010006_

Round 1

Reviewer 1 Report

Comments and Suggestions for Authors

Kansas City team proposed an interesting paper concerning TACE in HCC patients in US, and their readmission risk.

Paper is interesting, well-written.

I have juste minor questions:

- Is there influence of beads type on readmission?

- Just, remind in introduction post-embolization syndrome.

- Can you explain charges differences in the two populations?

Author Response

Thank you for your kind review and comments. Here is a point-by-point response to the comments in red text: 

- Is there influence of beads type on readmission? - We are not able to assess the types of beads from this database as this is not recorded or available for analysis.

- Just, remind in introduction post-embolization syndrome. - Post-embolization syndrome is mentioned as a complication in the introduction.

- Can you explain charges differences in the two populations? The difference in hospital charges between the two groups was not significant with a p value of 0.275

Reviewer 2 Report

Comments and Suggestions for Authors

Thank you for the opportunity to review your manuscript "Outcomes and Predictors of 30-Day re-admission in patients with HCC undergoing TACE between 2016-2019" where you analyze patients who undergo traditional/DEB TACE and found a readmission rate of 16% with weight loss and hepatic encephalopathy being risk factors for readmission. Comments on the manuscript are as follows.

1. The title is between 2016-2019 and the analysis of the NRD is from 2016-2018; is there a typo in the title?

2. Can the type of TACE be distinguished in the NRD? Patients who receive traditional TACE would often have a higher rate of side effects. If this can be distinguished, a supplementary analysis would be of interest.

3. Does the NRD report costs? Adding this information in would be helpful for economic analysis as charge does not reflect the real price of delivering care.

4. In Table 2, the n of each complication could be included. For the ones with 1 complication, that perhaps could be put as a supplemental table rather than the main one [suggestion, not strong].

5. Please check if it should be sex (biological) or gender (sociological) in the NRD; if it's gender, explicitly comment on this in the text. 

6. In table 3, peptic ulcer disease is misspelled in the footer.

Author Response

Thank you for your kind review and comments. Here is a point-by-point response (in red font)  to the comments:

1. The title is between 2016-2019 and the analysis of the NRD is from 2016-2018; is there a typo in the title? Type has been corrected to read 2018.

2. Can the type of TACE be distinguished in the NRD? Patients who receive traditional TACE would often have a higher rate of side effects. If this can be distinguished, a supplementary analysis would be of interest. We are not able to assess the types of beads from this database as this is not recorded or available for analysis.

3. Does the NRD report costs? Adding this information in would be helpful for economic analysis as charge does not reflect the real price of delivering care. NRD databse does not report the costs incurred by patients. We agree with the reviewer that charges do not adequately reflect the real price of delivering care.

4. In Table 2, the n of each complication could be included. For the ones with 1 complication, that perhaps could be put as a supplemental table rather than the main one [suggestion, not strong]. This has been verified and added.

5. Please check if it should be sex (biological) or gender (sociological) in the NRD; if it's gender, explicitly comment on this in the text. Thank you for this insightful comment. NRD database does report sex and not gender. Text has been modified to read sex.

6. In table 3, peptic ulcer disease is misspelled in the footer. This has been corrected, thank you!

Reviewer 3 Report

Comments and Suggestions for Authors

Thank you for the interesting study and for the findings. That will be very helpful for future researchers as well as for the clinicians.

Please mention the reason of taking TACE between 2016-2018 only, not up to the date or last year.

If possible, please prepare a graphical abstract for a clear and easier illustration.

Author Response

Thank you for your kind review and comments. Here is a point-by-point response (in red font) to the comments:

Please mention the reason of taking TACE between 2016-2018 only, not up to the date or last year.

- This was the available data accessible to the authors at the time of analysis.

If possible, please prepare a graphical abstract for a clear and easier illustration.

- As we are reporting only readmission rates, a graphical abstract would 

We are focused exclusively on presenting outcomes pertaining to 30-day readmissions, and due to the limited number of factors considered, creating a graphical abstract that accurately represents the data poses a considerable challenge.